# Habitat context affects sediment nitrogen burial by restored Eastern Oyster reefs

Anne Margaret H. Smiley[1,2*], F. Joel Fodrie[2], Jonathan H. Grabowski[3], Antonio B. Rodriguez[2], Suzanne P. Thompson[1,2], Michael F. Piehler[1,2]

1 Institute for the Environment, University of North Carolina, Chapel Hill, North Carolina, United States of America, 2 Department of Earth, Marine, and Environmental Sciences, University of North Carolina, Chapel Hill, North Carolina, United States of America, 3 Department of Marine and Environmental Sciences, Northeastern University, Boston, Massachusetts, Unites States of America

* ahsmiley@live.unc.edu

## Abstract

Oysters perform essential functions in estuarine environments. Reef restoration has recently become the subject of significant attention to reestablish populations after historic losses and to restore valuable ecosystem functions and services, including nitrogen removal. Nitrogen burial in oyster reef sediments may be an important nitrogen sink, but direct measurements are lacking. We assayed sediments from 11- to 14-year-old restored oyster reefs in three representative habitat contexts in a temperate estuary on the US Atlantic Coast. Elemental analysis of deep core sediments revealed that nitrogen burial rates ranged between 1.02 and 14.7 g N m$^{-2}$ y$^{-1}$ and generally scaled positively with reef relief and density. Intertidal flat reefs exhibited the greatest relief values, densities, and nitrogen burial rates. Subtidal flat reefs produced the lowest relief values and burial rates. Intertidal fringing reefs exhibited the lowest mean carbon:nitrogen ratio, 15.5 ± 1.3—burying proportionally more nitrogen than reefs in other habitat contexts. Using avoided cost methods, the value of nitrogen burial by oyster reefs in all habitat contexts ranged from $270 to $3,900 US dollars (USD) per hectare per year with an average of $1,700 USD per hectare per year. Integrating this figure into current estimates of nitrogen removal ecosystem services would increase the value 25–42%. Our findings suggest that specific site selection for restored and protected reefs can maximize nitrogen removal through multiple mechanisms, including burial. Providing empirical measurements of ecosystem function and estimates of economic value can inform site selection and design of restored oyster reefs to maintain water quality.

## Introduction

Oysters provide essential ecosystem services that directly benefit coastal communities by supporting economically important fisheries, reducing damage during storm events, and maintaining water quality [1–6]. Following historic losses—up to 85%

**Data availability statement:** All data files are available in the Carolina Digital Repository (https://cdr.lib.unc.edu/concern/data_sets/gq67k604s).

**Funding:** This research was supported by the UNC Institute for the Environment, the National Science Foundation (OCE-1233327, OCE-1155628, OCE-1635950), the United States Coastal Research Program (W912HZ-22-COO-11), and North Carolina Sea Grant (12-HCE-20). The funders had no role in study design, data collection and analysis, decision to publish, or preparation of the manuscript.

**Competing interests:** The authors have declared that no competing interests exist.

globally [7]—increased oyster reef restoration in recent decades has attempted to reestablish populations and restore valuable ecosystem services [8]. Importantly, oysters' ability to capture, produce, and retain sediments enable them to keep up with sea level rise [9,10], distinguishing reefs as intrinsically resilient nature-based solutions for coastal communities on the frontlines of change.

Passive accumulation of allogenic materials coupled with active organic matter generation through deposition of feces and pseudofeces link oyster accretion water quality. Physical and biogeochemical processes in the shallow sediments, or the taphonomically active zone (TAZ), and subsequent burial can influence availability of nitrogen for bloom-forming phytoplankton. Microbial decomposition of organic nitrogen produces inorganic, bioavailable forms of nitrogen that may be removed via bioassimilation [11–13], denitrification [3,14–17], or otherwise efflux out of the sediments [18,19]. Nitrogen compounds that evade these processes are buried below the TAZ, where they are no longer bioavailable. Burial in oyster reef sediments has been suggested as an important sink for nitrogen [20,21], but direct measurements of burial rates in these habitats are lacking.

Environmental conditions, such as subaerial exposure, flow velocity and adjacent habitats, can influence oyster reef structure and functioning [22,23]. Reef relief and vertical position in the water column can affect oyster recruitment [24–26] and potentially reef assimilation of nitrogen and/or production of nitrogen-rich biodeposits. Exposure to high flow velocities has a strong influence on reef size and complexity [27], and surface area for sediment capture. Proximity to other estuarine habitats, such as marshes, can influence sediment flux and composition [28], biogeochemical processing in the TAZ, and ultimately burial. A comprehensive understanding of environmental variables affecting ecological function is essential for effective oyster reef restoration.

Oyster restoration can be costly [29,30], and comprehensively accounting for ecosystem services provided by oyster reefs can help to quantify returns on investment and strategically design restoration projects [31,32]. Oysters have marketplace value through productive fisheries, but this alone does not capture the non-market value provided through additional ecosystem functions, such as nitrogen removal. Grabowski et al. [32] estimated that oyster reefs provide between $5,000 and $100,000 worth of ecosystem services per hectare annually, with nearly 40% of the total non-market value attributed to nitrogen removal on average. Notably, these figures do not include nitrogen removal through burial, highlighting the need to better quantify this process.

The objectives of this study were (1) to quantify sediment nitrogen content and burial rates in restored oyster reefs in varied habitat contexts in a temperate estuary on the US Atlantic Coast, and (2) determine whether restored oyster reefs bury nitrogen at economically and environmentally relevant rates. To do this, we assayed sediments from 11- to 14-year-old restored oyster reefs in three representative estuarine habitat contexts: intertidal flat, subtidal flat, and intertidal fringing reef. Positions within the tidal frame and relative to vegetation likely influence the capture, processing, and burial of nitrogen. We also calculated the value of nitrogen removal

from burial using an established nutrient credit trading system, and compared nitrogen burial rates to nitrogen loads from a municipal wastewater treatment facility. Though the capacity of restored oyster reefs to bury nitrogen has not been measured before in US Atlantic Coast estuaries, it is critical for a complete understanding of the ecosystem functioning and economic values of restored oyster reefs in particular and oyster reefs in general.

## Methods

### Study sites

In 2011, we sampled 20 experimental oyster reefs that were created in 1997 or 2000 in Back Sound, NC, USA (Fig 1) using approximately 60 bushels of oyster shell to achieve dimensions of roughly 5 x 3 x 0.30 m [33,34], representative of natural reef sizes in this region [27]. The reefs are scattered around Middle Marsh, a setting characterized by salt marshes, seagrass beds, and sand/mud flats, experiencing semi-diurnal tides that range approximately 0.9 m [9,28]. They are located within the Rachel Carson National Estuarine Research Reserve and have been protected from harvest since their construction. We sampled reefs across habitat contexts and inundation regimes, including intertidal flat (n = 7), subtidal flat (n = 3), and intertidal adjacent to saltmarsh hereafter referred to as intertidal fringing reef (n = 10). Reefs considered in a "flat" habitat context are within a mudflat or sandflat. The two westernmost subtidal flat reefs are seasonally surrounded by seagrass.

### Reef properties

**Relief.** We surveyed the reefs and surrounding areas using a Trimble Real Time Kinematic Global Positioning System mounted on a 2.0 m staff with an average vertical precision of 1.5 cm. Survey points were collected about every 50 cm along two reef-crossing transects. Transects across the patch reefs were oriented perpendicular to each other and crossed at the reef center. The transects across the fringing reefs were oriented parallel and perpendicular to the trend of the marsh shoreline and crossed near the edge of the salt marsh. The reefs are flat on top with steep sides; therefore, most of the survey points were obtained either from areas on top of the reef or on the adjacent sand flat. We defined reef relief as the difference between the average elevation of the reef top, based on 5–12 observations (± 1 SD), and the average elevation of the adjacent sandflat based on 4–10 observations (± 1 SD).

**Density.** To estimate oyster densities across each reef in which nitrogen burial estimates were made, we located two random 0.25-m$^2$ quadrats on intertidal flat and intertidal fringing reefs. The top 10 cm of material within each 0.25-m$^2$

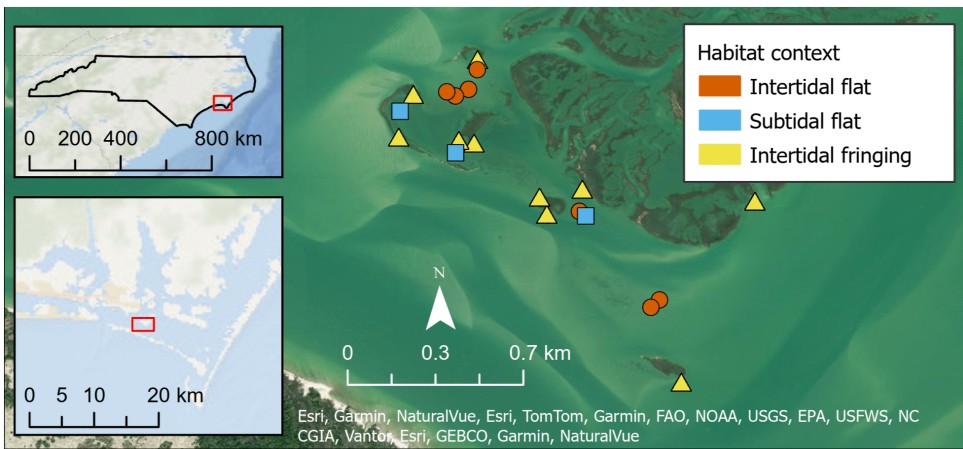

**Fig 1. Site map of 20 sampled oyster reefs in Middle Marsh, Back Sound, NC, USA.**

quadrat was excavated. Samples were sieved though a 1-mm mesh, and every live oyster was counted and measured to the nearest mm from the hinge to the leading growth margin. Density measurements were not collected for subtidal reefs as quadrat sampling could not be done in a manner consistent with sampling at intertidal reefs; including them would compromise comparability across habitat contexts.

**Sediment nitrogen and carbon contents.**  To quantify the nitrogen composition of reefs, we drove one 10 cm diameter aluminum pipe vertically through the X-Y center of each oyster reef using a gas-powered jack hammer. Cores sampled the entire reef structure (10-55 cm deep) and a few decimeters of the underlying substrate. Cores were sealed and transported to the Institute of Marine Sciences in Morehead City, NC, where they were stored upright in a 2 degree C walk-in freezer. Within a week from collection, cores were sectioned into 5 cm increments, which were then ground, fumed with 1NM HCl, and re-dried prior to induction in a Perkin Elmer CHN analyzer (Model 2400) to determine % nitrogen and % carbon (carbon data reported in Fodrie et al. 2017). Bulk-weight and % nitrogen data of sediments were combined to quantify nitrogen (by weight, g) in each core section (5 cm). Measurements of nitrogen were vertically integrated to produce estimates of total nitrogen, or lifetime burial, in reefs (g N m$^{-2}$). To calculate burial rates, we divided the overall weight of nitrogen within each down-core sample (5 cm increments, $i$) by the age of each experimental oyster reef (Eq. 1).

$$Burial\ rate = \frac{1}{Reef\ age} \sum_{i=1}^{n} Bulk\ weight_i * \frac{\%N_i}{100}$$

(1)

## Statistical methods

Several complimentary analyses were performed to explore patterns and correlates of nitrogen burial within restored experimental oyster reefs located on intertidal sand flats, subtidal sand flats, or adjacent to salt marshes within Middle Marsh.

First, we used linear regression analysis with data from all 20 experimental reefs to consider the relationship between total nitrogen content (lifetime burial) within oyster reefs (g N m$^{-2}$) and the vertical relief of each reef (i.e., upward accretion; in cm). Next, a polynomial regression was used to examine the link between live oyster density (individuals m$^{-2}$) and nitrogen burial rate (g N m$^{-2}$ yr$^{-1}$) at intertidal flat and fringing reefs. A second order fit was chosen following visual inspection of the X-Y scatterplot. Since density data were unavailable for subtidal sand flat reefs, they were excluded from this analysis. Reef-scale densities were based on the average of the two quadrats sampled from each reef. In these analyses, we considered nitrogen content and burial rates the dependent variables.

We utilized one-way ANOVAs to test for effects of habitat context on nitrogen content (g N m$^{-2}$) and reef densities (# m$^{-2}$) within the 11-to-14-year-old experimental oyster reefs. These data were tested for normality using Shapiro-Wilk tests across the main effect of habitat context, and no transformations were required to satisfy the assumptions of parametric analyses. Based on a significant main effect ($\alpha < 0.05$), we used Tukey HSD tests for post-hoc comparisons between treatments. Nitrogen burial rates (g N m$^{-2}$ y$^{-1}$), carbon:nitrogen (C:N) ratios, and relief (cm) data violated the assumption of normality; therefore, Kruskal Wallis and post hoc Dunn tests were used to compare means. Finally, the linear relationship between % carbon and % nitrogen within sediments was examined using regression analysis. For these analyses, we used the CHN outputs from each 5-cm core section as individual data points.

## Economic valuation and environmental relevance of nitrogen removal services

We extrapolated our measurements across reef area within Carteret County, NC according to NC Department of Marine Fisheries' Estuarine Benthic Habitat Mapping Program, Iteration 2, to evaluate prevalence of reefs in each habitat context and to estimate the amount and value of nitrogen burial at a local scale. Coastal counties in NC comply with the State's Coastal Area Management Act (15A North Carolina Administrative Code 07B), playing an important role in estuarine habitat management. We further rationalize a county-wide assessment due to the availability of data at this scale [35].

To estimate annual nitrogen removal rates by oyster reefs (kg N $y^{-1}$), burial rates for each habitat context were multiplied by the total reef area, which was determined considering only shell-bottom features. Features classified as "intertidal firm non-vegetated", or "intertidal hard non-vegetated" bottom were used to delineate intertidal sand flat reefs. Features labeled as "subtidal firm non-vegetated", or "subtidal hard non-vegetated" bottom were combined to identify subtidal sandflat reefs. Intertidal fringing reefs were delineated using features classified as "intertidal firm vegetated" or "intertidal hard vegetated" bottom. Substrates that were classified as "soft" were excluded from this spatial analysis. The habitat survey protocol does not distinguish between natural and restored reefs; however, Chambers et al. [36] found that restored intertidal reefs rapidly increase sediment nutrient concentrations and are comparable with older restored reefs and natural reefs within a year of construction. Therefore, we assumed that measurements collected from the restored reefs in Back Sound are representative of natural reefs that are closed to harvest. A limitation of this dataset, however, is that it does not distinguish between reefs that are protected/closed from harvest from those that are actively harvested.

We repeat the process for a subset of the Carteret County's estuarine waters in Calico Creek. Oyster reefs in Calico Creek are located downstream of Morehead City's wastewater treatment facility (NPDES permit #NC002661 or #NCG110110) and are closed to harvest due to chronically high concentrations of pollutants. Annual nitrogen burial rates were compared to dissolved inorganic nitrogen (DIN) inputs from this anthropogenic nitrogen point source. Annual DIN loads were obtained from the Environmental Protection Agency's Enforcement and Compliance History Online (ECHO) database (https://echo.epa.gov). The average annual loading rate was calculated by taking the average of values reported between 2013–2015 and 2020–2023—no values were reported from 2016–2019 in the ECHO database; therefore, those years were excluded from this analysis.

The economic value of nitrogen removal through burial by oyster reefs was calculated using rate schedules established in North Carolina's Nutrient Offset Program (15A North Carolina Administrative Code 2B.0703; Piehler & Smyth, 2011). The NC Department of Environmental Quality (DEQ) valued one kilogram of nitrogen removed at $26.39 in 2024 US dollars (USD) in the Tar-Pamlico basin. This figure was multiplied by annual nitrogen removal rates for each habitat context, which integrates the fine-scale nitrogen burial measurements and the spatial coverage of that type of reef.

## Results

### Sediment nitrogen across habitat contexts

Total nitrogen composition quantified for each reef revealed that lifetime nitrogen burial ranged from 13.3 to 162 g N $m^{-2}$ with a mean of 78.8 ± 9.6 g N $m^{-2}$ across habitat contexts. Total nitrogen content exhibited a significant positive linear relationship with reef relief ($R^2$ = 0.67, $p < 0.001$; Fig 2A). Nitrogen burial rates were calculated for each of the 20 restored oyster reefs and ranged between 1.03 and 14.7 g N $m^{-2}$ $y^{-1}$ with a mean of 6.42 ± 0.82 g N $m^{-2}$ $y^{-1}$. A significant parabolic relationship was identified between live oyster density and nitrogen burial rates in intertidal sand flat and fringing reefs ($R^2$ = 0.63, $p < 0.001$; Fig 2B). The second order polynomial fit predicted maximum burial rates to occur at reef densities of 3,000 live oysters $m^{-2}$.

Habitat context had a significant effect on reef relief, live oyster density, nitrogen composition, and nitrogen burial rates (Fig 3). Intertidal sand flat reefs had significantly greater mean relief (50.9 ± 4.1 cm) than either to subtidal flat or intertidal fringing reefs (16.7 ± 3.4 cm and 22.1 ± 3.9 cm, respectively; $p < 0.05$). Live oyster densities in intertidal flat reefs were 3497 ± 473 $m^{-2}$, which was significantly greater than those in intertidal fringing reefs (445 ± 65 $m^{-2}$; $p < 0.05$). Intertidal flat reefs also contained the greatest amount of nitrogen with 110 ± 9 g N $m^{-2}$, which was significantly higher than lifetime burial in subtidal sand flat reefs (38.0 ± 11.6 g N $m^{-2}$; $p < 0.05$). Lifetime burial in intertidal fringing reefs (69.4 ± 14.2 g N $m^{-2}$) was slightly lower than in intertidal sand flat reefs, but this difference was not statistically significant ($p = 0.08$). Nitrogen burial rates were significantly higher in intertidal sand flat reefs (9.03 ± 0.74 g N $m^{-2}$ $y^{-1}$) than in subtidal sand flat and intertidal fringing reefs (3.57 ± 1.06 g N $m^{-2}$ $y^{-1}$ and 5.45 ± 1.26 g N $m^{-2}$ $y^{-1}$, respectively; $p < 0.05$; Fig 3).

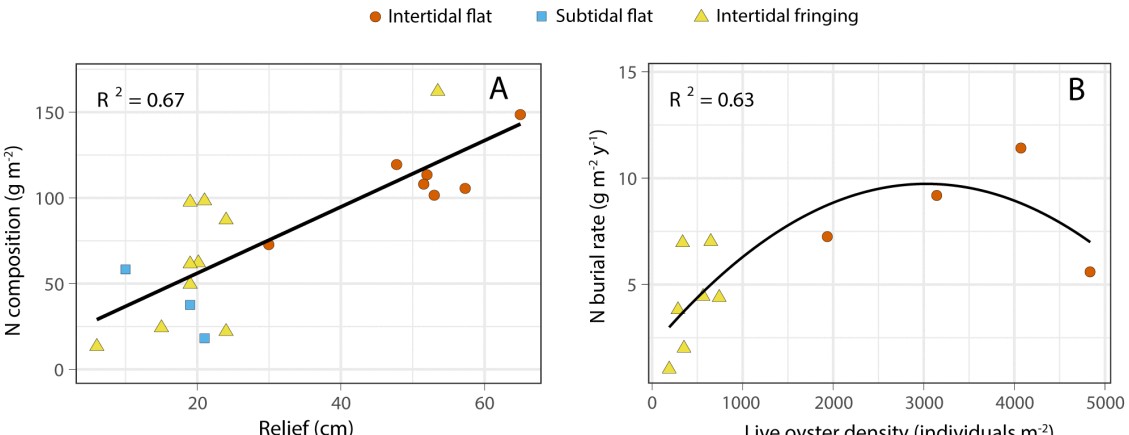

**Fig 2. Relationships between oyster-reef vertical relief (cm) and total nitrogen stored within oyster reefs (g N m$^{-2}$; A); and live oyster density (individuals m$^{-2}$) and annually averaged nitrogen burial rates by intertidal restored oyster reefs (g N m$^{-2}$ y$^{-1}$; B).**

For all restored oyster reefs, % carbon scaled positively with % nitrogen ($R^2 = 0.742$, $p < 0.05$; Fig 4), with intertidal sand flat reef sediments containing the highest amounts carbon and nitrogen compared to other reefs. Significant linear relationships between % carbon and % nitrogen in reef sediments were evident for each individual habitat context ($p < 0.05$). Average C:N ratios significantly differed across habitat contexts, with intertidal fringing reef sediments producing significantly lower values than those in either intertidal or subtidal sand flat reefs ($p < 0.05$; Table 1).

## Economic valuation of nitrogen burial

In Carteret County, there are approximately 274 hectares of intertidal flat oyster reefs, 2400 hectares of subtidal flat oyster reefs, and 166 hectares of intertidal fringing oyster reefs. Annually, Carteret County oyster reefs removed roughly 120,000 kg of nitrogen through burial valued at approximately $3,160,000 USD. Subtidal sand flat oysters comprise the largest portion at around $2,260,000 USD, followed by intertidal flat reefs at ~$653,000 USD and intertidal fringing reefs at ~$244,000 USD. Using the mean nitrogen burial rate across all habitat contexts, we estimated that the average annual value of nitrogen burial by oyster reefs is $1,720 USD per hectare.

We also report nitrogen burial and economic value of this ecosystem service provided by oysters in Morehead City, NC (Fig 5). This tidal creek hosts oyster reefs of all three habitat contexts investigated in this study and is affected by wastewater effluent, affording the opportunity to relate removal through burial to input from a nitrogen point source. In Calico Creek, there are roughly 28.9 hectares of oyster reefs comprised of 19.2 hectares of intertidal flat, 3.5 hectares of subtidal flat, and 6.2 hectares of intertidal fringing reefs. Collectively, these reefs remove approximately 2,200 kg N yr$^{-1}$ through burial. The Morehead City wastewater treatment facility adds roughly 10,900 kg DIN yr$^{-1}$ to Calico Creek; therefore, oysters in this system potentially remove 20% of the wastewater treatment facility's DIN load through burial.

## Discussion

This is the first study to directly measure nitrogen burial rates in oyster reefs in a temperate estuary on the US Atlantic Coast. Compared to other structured estuarine habitats in this ecoregion, like marshes and seagrasses, oyster reefs can bury nitrogen at considerably higher rates. Marshes in coastal Georgia, USA accumulated nitrogen at rates up to 6.9 g N m$^{-2}$ y$^{-1}$ [37,38]. Aoki et al. [39] reported nitrogen burial rates in restored seagrass meadows in South Bay, VA up to 3.5 g-N m$^{-2}$. The complexity/rugosity of oyster reefs compared to vegetated habitats may promote sediment capture and explain

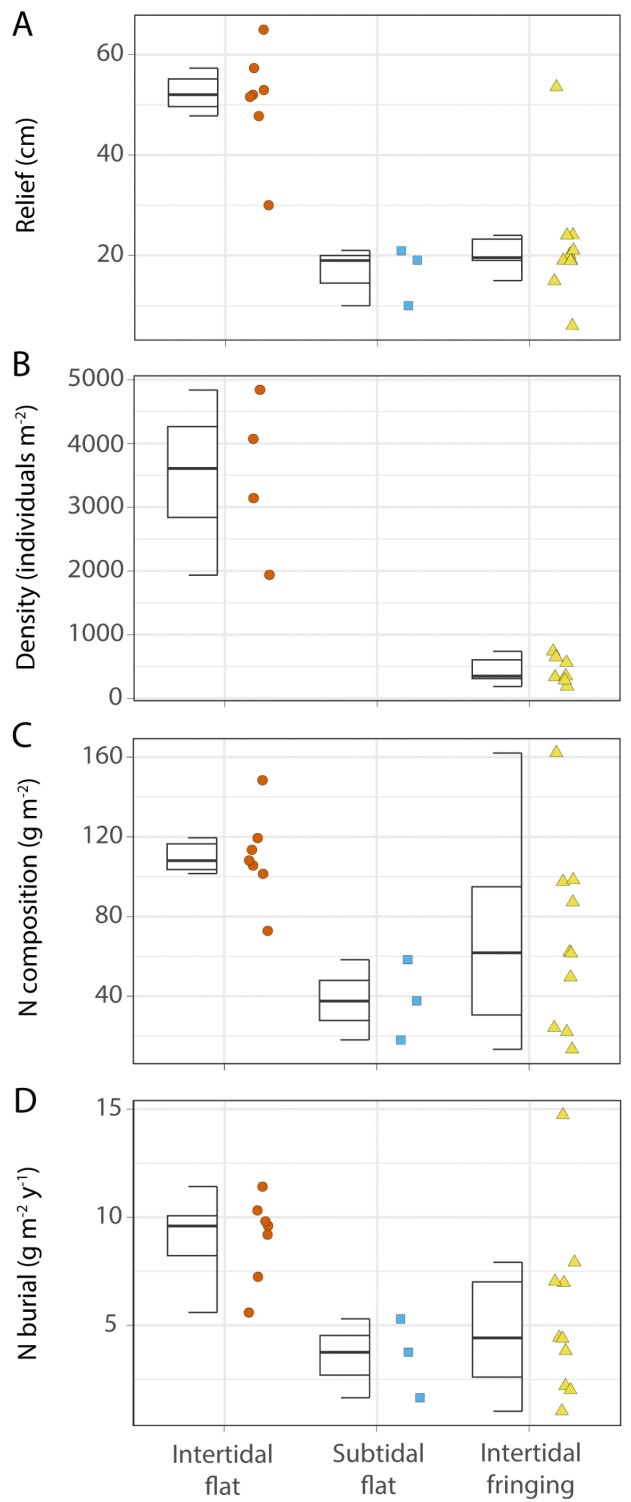

**Fig 3. Mean values across habitat contexts for relief (cm; A), density (individuals m$^{-2}$; B), N composition (g N m$^{-2}$; C), and burial rate (g N m$^{-2}$ y$^{-1}$; D).**

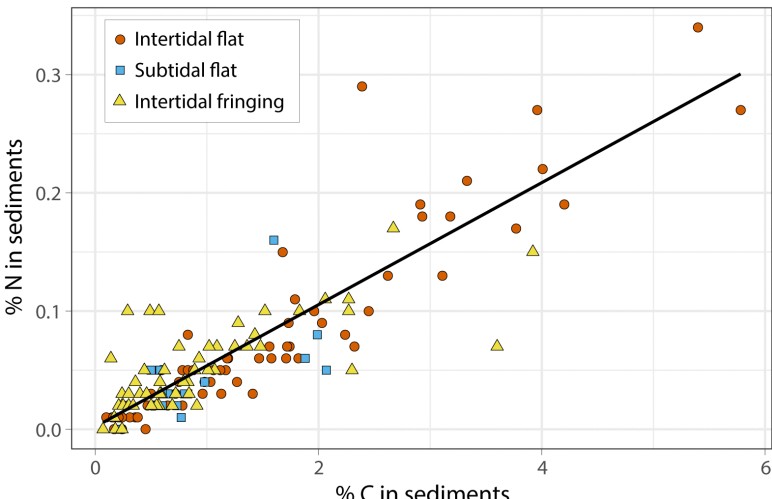

**Fig 4. Linear regression to compare % nitrogen and % carbon in oyster reef sediments for each habitat context.** Each datum represents a 5-cm vertical section of a core collected from oyster reefs. Analysis produced an $R^2$ value of 0.742.

**Table 1. Summary of mean C:N ratio across habitat contexts, calculated using values from each core section. Superscripts indicate statistical significance.**

| Habitat context | n | C:N ratio |
|---|---|---|
| Intertidal flat | 68 | 22.0±0.9[a] |
| Subtidal flat | 17 | 27.0±3.8[a] |
| Intertidal fringing | 54 | 18.5±1.3[b] |

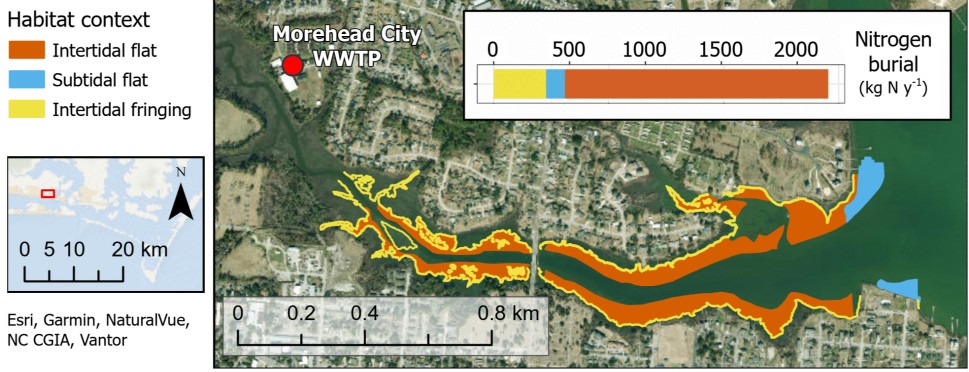

**Fig 5. Map of oyster reefs in Calico Creek, NC, USA, as delineated by the NC Department of Marine Fisheries.** Average nitrogen burial rates for each habitat context were multiplied by the area of respective reefs to estimate annual removal and value of the ecosystem service.

why the upper threshold measured in the restored oyster reefs was more than double the values recorded in the marsh and more than four times those in seagrasses. Thus, to maximize nitrogen burial in estuaries, restoration of oyster reefs should be prioritized over other structured habitats.

Nitrogen burial in oyster reefs is an important sink, comparable with other nitrogen removal mechanisms. Mean nitrogen burial measured in this study falls within the range reported by Smyth et al. [40] for average annual denitrification

rates in Bogue Sound—5−7 g N m$^{-2}$ y$^{-1}$. Compared with bioassimilation of nitrogen into oyster tissues and subsequent harvest, nitrogen removal via burial can be up to five times greater; Beseres Pollack et al. [21] estimated that oyster reefs in the Mission-Aransas Estuary can remove roughly 1.2 g N m$^{-2}$ y$^{-1}$ through harvest. Accounting for nitrogen removal via burial will significantly increase previous value estimates of nitrogen removal ecosystem services provided by oyster reefs.

## Environmental context affects nitrogen burial

Nitrogen burial was strongly correlated with oyster reef structure, with lifetime burial scaling positively with relief and burial rates exhibiting a parabolic relationship with density. Physical-biological feedbacks link environmental variables such as flow velocity and tidal range, to reef morphology. For example, exposure to high flow velocities can also increase reef complexity [23], which may enhance sediment capture and retention and promote vertical reef building [41]. Higher energy can also mix the water column and increase the food supply to oysters to enhance their growth [42,43] and production of biodeposits.

Regional differences in tidal influence may explain differences in morphology, and consequently nitrogen burial between Atlantic Coast oyster reefs included in this study and Gulf Coast oyster reefs. Westbrook et al. [13] examined oyster reefs in coastal Louisiana (LA), which are generally more scattered and sparser compared to Atlantic Coast reefs, configured in denser more well-defined patches. Oyster reefs in the LA study [13] experience diurnal tides ranging approximately 0.3 m and exhibited densities an order of magnitude lower and nitrogen burial rates over six times lower than those measured in Back Sound, NC where semidiurnal tides range roughly 1 m. These differences highlight the importance of regional context in restoration outcomes.

Differences in harvest pressure may also explain differences in reef morphology and nitrogen burial rates in LA and NC oyster reefs. Nitrogen burial was strongly correlated with reef relief and generally scaled positively with density; thus, active removal of individuals may hamper the capacity to bury nitrogen. The LA reefs examined in the Westbrook et al. study [13] are subject to active harvest, potentially explaining the low nitrogen burial rates compared to the protected reefs in Middle Marsh, NC. This information has implications for reefs that are closed to harvest for conservation, fishery repopulation, and/or chronic exposure to pollutants.

At the local scale, position within the tidal frame affected reef structure and nitrogen burial—intertidal flat reefs had greater relief and buried nitrogen at significantly higher rates than subtidal flat reefs. Compared to subtidal settings, the intertidal position creates conditions more favorable for oyster growth, leading to increased production of biogenic sediments and nitrogen burial [44]. For example, some percentage of aerial exposure can reduce biofouling and predation from pelagic species [45]. Ridge et al. [46] identified the optimal growth zone (OGZ) for oysters in Back Sound, NC as the height at which the reef is subaerial 20–40% of the time. The intertidal position may also enhance the capture of allogenic sediments, as higher relief may provide a greater surface area that intercepts waters that carry sediments.

Despite similar positions within the tidal frame, intertidal salt marsh fringing oyster reefs exhibited different structural characteristics and lower nitrogen burial rates compared to intertidal sand flat reefs. Fringing reefs are typically positioned near the lower boundary of the OGZ [47], explaining the relatively low relief values and densities compared to intertidal flat reefs. Adjacent salt marsh may also slow flow rates on the landward edge of fringing reefs and attenuate the flow of seston, producing less favorable conditions for biogenic sediment production and vertical accretion. Moreover, the parabolic relationship observed between density and burial rates may reflect differences between fringing and flat habitat contexts. Intertidal fringing reefs accumulate relatively high amounts of nitrogen at relatively low densities. It may be that proximity to salt marsh simultaneously reduces flow rates and supplies additional sediments, allowing material accumulation in the interstitial spaces of the fringing reefs. In contrast, intertidal sand flat reefs exhibit high densities and nitrogen burial rates plateau. In this setting, erosional and depositional forces may be balanced differently such that oyster recruits, rather than sediments, fill the interstitial space [46].

Proximity to vegetation also significantly affected sediment contents in the restored oyster reefs. Fringing reef sediments exhibited lower C:N ratios than other habitat contexts, meaning they bury proportionally more nitrogen. Differences

in sediment sources and biogeochemical processes in the TAZ likely explain the observed differences in sediment contents. Westbrook et al. [13] discuss elevated nitrogen concentrations in shallow water reefs compared to deep-water reefs in LA, citing inputs from productive marshes. It is possible that in Back Sound, NC, marshes augment the nitrogen supply in their adjacent fringing reefs, compared to habitat contexts more distal to vegetation. Differences in flow regime and organic matter source/flux between intertidal fringing and flat reefs may indirectly affect sediment C:N ratios by affecting the phytoplankton availability in seston [48]. Wave baffling and slowed flow rates on the landward edge of fringing oyster reefs [47] may favor growth of benthic microalgae [49], which has been correlated with high biodeposit quality (lower C:N ratio; [50]). Another possible explanation for different C:N ratios in intertidal fringing versus flat reef habitat contexts is nitrogen removal through additional biogeochemical processes, such as denitrification. Smyth et al. [14] reported oyster-enhanced denitrification in estuarine flat sediments, but not in salt marsh sediments. It's possible that nitrogen removal in the surrounding sand flat sediments (via coupled nitrification-denitrification) reduces the proportion of nitrogen in accumulated allogenic sediments and results in higher C:N ratios relative to fringing reefs.

Another important consideration is site-specific water chemistry, particularly ambient nitrate concentrations, as they may affect processes in the TAZ and ultimately the nitrogen pool available for burial [51]. Smyth et al. [14] showed that oyster-mediated denitrification was significantly higher under elevated nitrate conditions compared to ambient concentrations. This is likely due to the increased contribution of direct denitrification, in which nitrate is extracted from the water column, contrasting coupled nitrification-denitrification in which nitrate is supplied by the sediments [52]. Chronically nitrate-enriched environments, such as waters downstream of a wastewater treatment facility [53], may provide conditions that favor nitrogen removal from the water column via direct denitrification, allowing a larger pool of sediment nitrogen to be buried. This interpretation supports spatially targeting restoration sites can maximize nitrogen removal via multiple mechanisms.

## Opportunities to inform restoration practices

Restoration costs are highly variable, ranging between $3,820 and $2,180,000 with an average cost of $299,999 USD per hectare [8] depending on project site, scale, and substrate [8,29,54,55]. This variability paired with an incomplete understanding of ecosystem value makes estimating returns on investment difficult. Our measurements of nitrogen burial contribute to a more comprehensive understanding of ecosystem function and value. We valued nitrogen burial by oyster reefs at approximately $1,700 USD per hectare per year. While this figure may only be 0.05% of average restoration costs, annual returns are cumulative—over 20 years, a restored reef may recover 10% of restoration costs—and contribute to the larger supply of provisionary, regulatory, and cultural ecosystem services. Grabowski et al. [32] estimated the annual value of nitrogen removal services—including denitrification and phytoplankton consumption— from oysters to be between $4,050 and $6,716 per hectare per year without accounting for removal via burial. Integrating our measurements of nitrogen burial could increase this figure by 25–42%.

The results of this study revealed that oyster reef restoration can be an effective strategy to reduce water column nitrogen through burial and should include consideration of habitat context. In Back Sound, NC, intertidal oyster reefs buried nitrogen at the highest rates and should be prioritized in restoration efforts and considered in concert with denitrification potential when the goal is nitrogen removal. Oyster reef restoration/construction plans should consider relief, ensuring that new reefs exist within the OGZ to promote sediment accumulation and resilience to sea level rise. Subtidal reefs produced the lowest average nitrogen burial rate of the three habitat contexts and likely exists below a critical exposure boundary under which growth rates will not keep up with sea level rise [46]. However, of the three habitat contexts evaluated in this study, subtidal reefs are the most abundant in Carteret County and provide the greatest overall economic value.

In addition to relief, restoration projects should also consider oyster densities to maximize water quality benefits. Intertidal oyster reefs (flat and fringing) exhibited a parabolic relationship between density and nitrogen burial, with a maximum occurring at 3,000 individuals m$^{-2}$ and Smyth et al. [14] identified a threshold of approximately 2,400 individuals m$^{-2}$

for maximum denitrification. Dame et al. [56] reported higher growth and recruitment rates in East Coast creeks where oysters had been removed. Oyster reefs continue to grow and remove nitrogen through burial and denitrification past these thresholds, but removal rates are slightly reduced. It's possible that these reefs exhibit a carrying capacity above which competition for space and nutrients limits growth and consequently production of nitrogen-rich biodeposits. Creating conditions that would enhance growth rates could also increase nitrogen removal through bioassimilation and extraction [12,57].

Living shoreline construction is a common restoration practice and adaptive management strategy that typically include an intertidal fringing oyster reef adjacent to a marsh [58]. We found that these types of reefs bury nitrogen rapidly at relatively low densities and bury proportionally more nitrogen than reefs in other habitat contexts. Onorevole et al. [59] demonstrated that living shoreline habitats in Bogue Sound, NC effectively remove nitrogen via denitrification at relatively young ages. Moreover, fringing reefs and their adjacent salt marshes each bolster ecosystem functions in the other. Examples include increased denitrification in marsh sediments under nitrate enriched conditions [40] and enhanced carbon burial in marshes when oysters are present [28]. Furthermore, their position at the land water margin physically armors the shoreline to protect against erosion during storms [1,60,61].

These data highlight how managing restoration schemes across multiple ecosystem services requires consideration. For instance, Fodrie et al. [62] showed that in terms of carbon removal, intertidal (flat) reefs are the 'worst' option–subtidal could be preferrable as service provider. In terms of juvenile fishes, Grabowski et al. [33] and Zeigler et al. [63] showed that intertidal flat reefs augment densities, but for larger sub-adult and adult fishes, saltmarsh fringing appear to be more valuable, possibly due to marsh-reef connectivity [45]. Optimal reef design is, thus, ultimately dependent on the restoration objectives, with these findings advancing an understanding of oyster reef-driven processes to achieve those objectives. By identifying environmental drivers of nitrogen burial, this work can inform effective coastal habitat and nutrient management to maximize water quality benefits.

## Acknowledgments

We would like to thank the field and laboratory teams that were instrumental in restored reef construction as well as sample collection and processing.

## Author contributions

**Conceptualization:** F. Joel Fodrie, Jonathan H. Grabowski, Antonio B. Rodriguez, Suzanne P. Thompson, Michael F. Piehler.

**Data curation:** Anne Margaret H. Smiley.

**Formal analysis:** Anne Margaret H. Smiley.

**Funding acquisition:** F. Joel Fodrie, Jonathan H. Grabowski, Antonio B. Rodriguez, Michael F. Piehler.

**Investigation:** F. Joel Fodrie, Jonathan H. Grabowski, Antonio B. Rodriguez, Suzanne P. Thompson, Michael F. Piehler.

**Methodology:** F. Joel Fodrie, Jonathan H. Grabowski, Antonio B. Rodriguez, Suzanne P. Thompson, Michael F. Piehler.

**Project administration:** F. Joel Fodrie, Jonathan H. Grabowski, Antonio B. Rodriguez, Suzanne P. Thompson, Michael F. Piehler.

**Supervision:** F. Joel Fodrie, Jonathan H. Grabowski, Antonio B. Rodriguez, Suzanne P. Thompson, Michael F. Piehler.

**Visualization:** Anne Margaret H. Smiley.

**Writing – original draft:** Anne Margaret H. Smiley.

**Writing – review & editing:** Anne Margaret H. Smiley, F. Joel Fodrie, Jonathan H. Grabowski, Antonio B. Rodriguez, Suzanne P. Thompson, Michael F. Piehler.

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
