## [Decision Letter · Decision Letter 0]

16 Dec 2025

PONE-D-25-25624
Habitat context affects sediment nitrogen burial by restored Eastern Oyster reefs
PLOS One

Dear Dr. Smiley,

Thank you for submitting your manuscript to PLOS ONE. After careful consideration, we feel that it has merit but does not fully meet PLOS ONE’s publication criteria as it currently stands. Therefore, we invite you to submit a revised version of the manuscript that addresses the points raised during the review process.

We look forward to receiving your revised manuscript.

Kind regards,

Nadeem Nazurally, Ph.D

Academic Editor

PLOS One

Journal Requirements:

This research was supported by the UNC Institute for the Environment, the National Science Foundation (OCE-1233327, OCE-1155628, OCE-1635950), the United States Coastal Research Program (W912HZ-22-COO-11), and North Carolina Sea Grant (12-HCE-20).

This research was supported by the UNC Institute for the Environment, the National Science Foundation (OCE-1233327, OCE-1155628, OCE-1635950), the United States Coastal Research Program (W912HZ-22-COO-11), and North Carolina Sea Grant (12-HCE-20).

This research was supported by the UNC Institute for the Environment, the National Science Foundation (OCE-1233327, OCE-1155628, OCE-1635950), the United States Coastal Research Program (W912HZ-22-COO-11), and North Carolina Sea Grant (12-HCE-20).

5. Please amend the manuscript submission data (via Edit Submission) to include author F. Joel Fodrie

6. Please amend your authorship list in your manuscript file to include author Joel Fodrie

7. Please include a caption for figure 5.

8. We note that Figures 1 and 5 in your submission contain map/satellite images which may be copyrighted. All PLOS content is published under the Creative Commons Attribution License (CC BY 4.0), which means that the manuscript, images, and Supporting Information files will be freely available online, and any third party is permitted to access, download, copy, distribute, and use these materials in any way, even commercially, with proper attribution. For these reasons, we cannot publish previously copyrighted maps or satellite images created using proprietary data, such as Google software (Google Maps, Street View, and Earth). For more information, see our copyright guidelines: http://journals.plos.org/plosone/s/licenses-and-copyright.

a. You may seek permission from the original copyright holder of Figures 1 and 5 to publish the content specifically under the CC BY 4.0 license.

Reviewers' comments:

Reviewer's Responses to Questions

**Comments to the Author**

1. Is the manuscript technically sound, and do the data support the conclusions?

Reviewer #1: Yes

Reviewer #2: Yes

2. Has the statistical analysis been performed appropriately and rigorously?

Reviewer #1: Yes

Reviewer #2: Yes

3. Have the authors made all data underlying the findings in their manuscript fully available?

Reviewer #1: Yes

Reviewer #2: Yes

4. Is the manuscript presented in an intelligible fashion and written in standard English?

Reviewer #1: Yes

Reviewer #2: Yes

5. Review Comments to the Author

Reviewer #1: The manuscript presents a rigorous and timely study quantifying nitrogen burial rates across different habitat contexts in restored oyster reefs. It fills a key knowledge gap by directly measuring an overlooked process in nutrient cycling and ecosystem functioning. The design is robust, analyses are appropriate, and results are well supported. The discussion effectively integrates regional perspectives but could be more concise, particularly when comparing burial with denitrification. The manuscript is clearly written and scientifically sound, requiring only minor editorial refinement. Overall, it is a strong, policy-relevant contribution linking biogeochemical research, restoration practices, and ecosystem service valuation.

Reviewer #2: The study is well-structured and highlights meaningful differences across habitat contexts. However, some interpretations seem stronger than the supporting data allow, and a few assumptions—particularly around extrapolation and habitat equivalency require clearer justification or acknowledgment as limitations. Improving clarity in the methods, moderating broad claims, and more explicitly discussing uncertainty would strengthen the overall findings and their applicability to restoration planning. The author should also use more recent references.

6. PLOS authors have the option to publish the peer review history of their article (what does this mean?). If published, this will include your full peer review and any attached files.

Reviewer #1: No

Reviewer #2: No

---

## [Author Response · Author response to Decision Letter 1]

13 Feb 2026

We would like to thank reviewers and editors for taking the time to read this manuscript and provide thoughtful feedback. Addressing these comments has improved the quality and clarity of this work. We have slightly restructured the Introduction and Discussion sections to improve the logical flow of themes and ideas. To do this, we combined the previously separate regional- and local-scale environmental context paragraphs/sections into one cohesive segment that relates environmental variability to structural differences in oyster reefs, then translated those structural differences to differences in functioning (nitrogen burial). We also provide additional methodological details as well as further explanation regarding spatial extrapolation, being more explicit about assumptions made and limitations of the methods. We include line-by-line response to specific comments below.

Responses to specific editor comments:

Response: Thank you, we have adjusted the formatting of the manuscript to adhere to PLoS ONE’s style requirements.

This research was supported by the UNC Institute for the Environment, the National Science Foundation (OCE-1233327, OCE-1155628, OCE-1635950), the United States Coastal Research Program (W912HZ-22-COO-11), and North Carolina Sea Grant (12-HCE-20).

Response: Thank you. Please add the statement "The funders had no role in study design, data collection and analysis, decision to publish, or preparation of the manuscript."

This research was supported by the UNC Institute for the Environment, the National Science Foundation (OCE-1233327, OCE-1155628, OCE-1635950), the United States Coastal Research Program (W912HZ-22-COO-11), and North Carolina Sea Grant (12-HCE-20).

This research was supported by the UNC Institute for the Environment, the National Science Foundation (OCE-1233327, OCE-1155628, OCE-1635950), the United States Coastal Research Program (W912HZ-22-COO-11), and North Carolina Sea Grant (12-HCE-20).

Response: Thank you. We have removed all funding-related language from the manuscript. We have added the following text to the Acknowledgements section (lines 404-405): “We would like to think the field and laboratory teams that were instrumental in restored reef construction as well as sample collection and processing.”

Response: Thank you. We have made all raw data included in our analyses available in a public repository hosted by the University of North Carolina at Chapel Hill at the following URL: https://cdr.lib.unc.edu/concern/data_sets/gq67k604s

5. Please amend the manuscript submission data (via Edit Submission) to include author F. Joel Fodrie

Response: Thank you. The list of authors includes F. Joel Fodrie as second author.

6. Please amend your authorship list in your manuscript file to include author Joel Fodrie

Response: Thank you. The list of authors includes F. Joel Fodrie as second author.

7. Please include a caption for figure 5.

Response: Thank you. the Caption for figure 5 is included and reads: “Fig 5. Map of oyster reefs in Calico Creek, NC, USA, as delineated by the NC Department of Marine Fisheries. Average nitrogen burial rates for each habitat context were multiplied by the area of respective reefs to estimate annual removal and value of the ecosystem service.”

8. We note that Figures 1 and 5 in your submission contain map/satellite images which may be copyrighted. All PLOS content is published under the Creative Commons Attribution License (CC BY 4.0), which means that the manuscript, images, and Supporting Information files will be freely available online, and any third party is permitted to access, download, copy, distribute, and use these materials in any way, even commercially, with proper attribution. For these reasons, we cannot publish previously copyrighted maps or satellite images created using proprietary data, such as Google software (Google Maps, Street View, and Earth). For more information, see our copyright guidelines: http://journals.plos.org/plosone/s/licenses-and-copyright.

Response: Thank you. Both figures were created using stock basemaps in ArcGIS Pro, and have been updated to include service layer credits.

Response: Thank you. The Reviewers did not recommend specific references. We did however, include additional, more recent citations upon Reviewer recommendation.

Response: Thank you. We have checked that all published work mentioned in the text are included in the references and properly formatted.

Responses to specific reviewer comments:

Line 18: Briefly mention the methods used

Response: Thank you for this suggestion, this change has been made. Lines 17-18 in the abstract now read: “Elemental analysis of deep-core sediments revealed that nitrogen burial rates ranged between…”

Line 21: How much? (in reference to C:N ratios)

Response: Thank you, this addition has been made. Line 21 in the abstract now reads: “Intertidal fringing reefs exhibited the lowest mean C:N ratio, 18.5 ± 1.3—burying proportionally more nitrogen than reefs in other habitat contexts”

Line 31: More recent citations need to be used, where possible. Generally, the introduction is too long and unfocused. Streamline by grouping content into 3–4 focused themes, directly connected to the study. Consolidate repetitive concepts for a better flow, such as ecosystem services and reef restoration. Same concepts are repeated multiple times at different points in the intro. Logical flow leading to the study objectives is needed.

Response: Thank you for this feedback. We have streamlined the introduction and improved connecting sentences to improve flow, highlight fewer main ideas, and convey the following logical progression:

1) Importance of oysters and potential for resilient nature-based solutions [due to their ability to capture and retain sediments]

2) Overview of nitrogen processing in oyster reef sediments, highlighting poor understanding of burial

3) Environmental variables that influence reef structure and function, necessary information for effective restoration

4) Quantifying value of ecosystem services can inform cost-effective restoration

5) Overview of objectives/study.

The following specific changes have been made to the introduction:

Lines 33-34: Removed “contributions to ecosystem function through the provision of habitat and nursery areas, sediment stabilization, and nutrient cycling. These functions provide”

Lines 35-37: Rearranged this sentence to read “following historical losses—up to 85% globally (Beck et al., 2011)—increased oyster reef restoration in recent decades has attempted to reestablish populations and valuable ecosystem services (Bersoza Hernandez et al., 2018).”

Lines 37-39: Shortened and rearranged this sentence to read “Importantly, oysters’ ability to capture, produce, and retain sediments enable them to keep pace with sea level rise…”

Line 39: Added a broader context statement “…distinguishing reefs as intrinsically resilient nature-based solutions for coastal communities on the frontlines of change.”

Lines 41-43: Shortened this sentence to read “…deposition of feces and pseudofeces link oyster reef accretion to water quality.”

Line 47-48: Changed “could be an important sink” to “has been suggested as an important sink”.

Lines 50-58: Combined what were previously two separate paragraphs on regional and local environmental variables to broadly encompass environmental variables that affect reef properties and processes. We also removed the section about physicochemical characteristics of the water column to be more concise and added a closing sentence to emphasize the importance of understanding environmental controls on nitrogen processing. The single paragraph now reads:

“Environmental conditions, such as subaerial exposure, flow velocity, and adjacent habitats (Byers et al., 2015; Keller et al., 2019) can influence oyster reef properties and functioning. Reef relief and vertical position in the water column can affect oyster recruitment (O'Beirn et al., 2000; Perog et al., 2023; Schulte et al., 2009) and potentially reef assimilation of nitrogen and/or production of nitrogen-rich biodeposits. Exposure to high flow velocities has a strong influence on reef size and complexity (Bahr & Lanier, 1981), and surface area for sediment capture. Proximity to other estuarine habitats, such as marshes, can influence sediment flux and composition (Ridge et al., 2017), biogeochemical processing in the TAZ, and ultimately burial. A comprehensive understanding of environmental variables affecting ecological function is essential for effective oyster reef restoration.”

Lines 65-67: Changed text to emphasize the need for this study. It now reads “Notably, these figures do not include nitrogen removal through burial, highlighting the need to better quantify this process.”

Line 72: Changed “3” to “three”

Line 33: Reference to this statement.

Response: Thank you, the sentence focused on ecosystem function has been removed, but the sentence focused on ecosystem services and the associated references remain.

Line 54: This paragraph is dense and confusing. Break into shorter sentences and clearly explain why burial is distinct from other pathways and why it matters.

Response: Thank you for this suggestion. We maintain that a brief overview of nitrogen cycling in oyster reef sediments is important context, as we draw comparisons to additional nitrogen processes in the discussion. We distinguish nitrogen burial as the terminal process that is poorly understood compared to processes in the dynamic taphonomically active zone that are well characterized in comparison. To highlight this knowledge gap, this paragraph now ends with “Burial is oyster reef sediments has been suggested as an important sink for nitrogen (Beseres Pollack et al., 2013; Newell et al., 2005), but direct measurements of burial rates in these habitats are lacking.” (lines 47-49)

Lines 90-91: Add a bridging statement explaining why these habitat contexts are expected to differ in nitrogen burial.

Response: Thank you for this suggestion. We added “Positions within the tidal frame and relative to vegetation likely influence the capture, processing, and burial of nitrogen.”

Line 128: Provide clear explanation why. (in refence to lack of subtidal density data)

Response: Thank you for this comment. We provided this explanation on lines 111-113: “Density measurements were not collected on subtidal reefs as quadrat sampling could not be done in a manner consistent with sampling at intertidal reefs; including them would compromise comparability across habitat contexts.”

Line 136: Exact number of cores per reef? How cores were handled (sealed? frozen? refrigerated?)

Response: Thank you for this comment, we agree adding this information would improve clarity and reproducibility. On line 115, we specified that one sediment core was collected from the X-Y center of each reef. We also added to lines 118-122 “Cores were sealed and transported to the Institute of Marine Sciences in Morehead City, NC, where they were stored upright in a 2 degree C walk-in freezer. Within a week from collection, cores were sectioned into 5-cm increments, which were then ground, fumed…”

Line 141: Specify equation

Response: Thank you. We have specified the equation as:

Burial rate= 1/(Reef age) ∑_(i=1)^n▒〖〖Bulk weight〗_i*〖%N〗_i/100〗

Line 165: Extrapolating burial rates to all county reefs assumes restored and natural reefs are equivalent, but the mapping data do not distinguish them. This may bias large-scale estimates and should be acknowledged or justified.

Response: Thank you. We agree that it’s important to acknowledge limitations associated with scaling site-level data to larger areas. In this paragraph (line 170), we commented on the lack of distinction between natural and restored reefs, citing Chambers et al. (2017), which concludes that within one year restored reefs reach sediment nutrient concentrations comparable to natural reefs. Therefore, we assume our data from restored reefs are representative of rates in natural reefs. We did clarify at the end of this paragraph (lines 174-175) that a limitation of this dataset “is that it does not distinguish between reefs that are protected or closed to harvest from those that are actively harvested.”

Line 228: The results present numerous pairwise comparisons but occasionally lack post-hoc p-values or effect sizes, making the interpretation of “significant” vs. “not significant” differences harder to follow.

Response: Thank you for pointing this out. We have added R2 and p-values throughout the results section to improve consistency and clarity.

Line 249: Relying on mean burial rates aggregated across habitats, may mask large differences betwee

---

## [Editor Report · Decision Letter 1]

18 Feb 2026

Habitat context affects sediment nitrogen burial by restored Eastern Oyster reefs

PONE-D-25-25624R1

Dear Dr. Smiley,

We’re pleased to inform you that your manuscript has been judged scientifically suitable for publication and will be formally accepted for publication once it meets all outstanding technical requirements.

Kind regards,

Nadeem Nazurally, Ph.D

Academic Editor

PLOS One
---

## [Editor Report · Acceptance letter]

PONE-D-25-25624R1

PLOS One

Dear Dr. Smiley,

I'm pleased to inform you that your manuscript has been deemed suitable for publication in PLOS One. Congratulations! Your manuscript is now being handed over to our production team.

Kind regards,

on behalf of

Dr. Nadeem Nazurally

Academic Editor

PLOS One